# Quality Change of Citri Reticulatae Pericarpium (Pericarps of *Citrus reticulata* ‘Chachi’) During Storage and Its Sex-Based In Vitro Digestive Performance

**DOI:** 10.3390/foods13223671

**Published:** 2024-11-18

**Authors:** Peirong Yu, Yuying Zeng, Chunyu Li, Bixia Qiu, Yuan Shi, Qixi He, Uri Lesmes, Yigal Achmon

**Affiliations:** 1Department of Biotechnology and Food Engineering, Technion-Israel Institute of Technology, Haifa 3200003, Israel; yu.peirong@gtiit.edu.cn (P.Y.); zeng.yuying@gtiit.edu.cn (Y.Z.); li.chunyu@gtiit.edu.cn (C.L.); bixia.qiu@gtiit.edu.cn (B.Q.); 2Department of Biotechnology and Food Engineering, Guangdong Technion-Israel Institute of Technology, GTIIT, 241 Daxue Road, Shantou 515063, China; shi05911@alumni.gtiit.edu.cn (Y.S.); he08204@gtiit.edu.cn (Q.H.); 3Guangdong Provincial Key Laboratory of Materials and Technologies for Energy Conversion, Guangdong Technion-Israel Institute of Technology, Shantou 515063, China

**Keywords:** Citri Reticulatae Pericarpium, *Citrus reticulata* ‘Chachi’, shelf life, volatile organic compounds, flavonoids, in vitro digestion models, bioaccessibility, consumer sex

## Abstract

Citri Reticulatae Pericarpium (CRP), particularly including the pericarp of *Citrus reticulata* ‘Chachi’ (GCP), has been widely used as a food, a dietary supplement, and traditional Chinese medicine. Despite the widespread use of traditional foods, there is limited evidence regarding the precise relationships between storage conditions, aging duration, and the digestive performance of CRP. In this study, the aim was to investigate the impact of the storage conditions on the quality of aged GCP during shelf life and to evaluate the subsequent digestive performance of corresponding GCP decoctions. Respiration in GCP was monitored by measuring oxygen (O_2_), carbon dioxide (CO_2_), and methane (CH_4_) gases throughout the storage simulation, with O_2_ and CO_2_ validated as prospective safety measures. Five flavonoids (hesperidin, didymin, nobiletin, tangeretin, and 3,5,6,7,8,3′,4′-heptamethoxyflavone) were determined as quality indicators, and their contents were significantly affected by the duration of the storage simulation and the aging periods of GCP. Our study also found that temperature and humidity significantly affected the volatile organic compounds (VOCs) emission from GCP. Eighteen compounds were proposed to show potential as descriptive measures of aging periods while eight compounds were proposed as potential indicators to discriminate among the spoilage level. Furthermore, the bioaccessibility of hesperidin ranged from ~30% to ~50% and was not significantly affected by the GCP’s aging time nor the consumer’s sex (*p* < 0.05). This study presents evidence for the future control of the quality of GCP and its digestive performance in males and females.

## 1. Introduction

Traditional foods have been consumed over extended periods, but the scientific knowledge about their quality and health impacts might remain limited. Currently, the intersection of traditional foods and human health gains great interest and requires systematic elucidation of their chemical composition and digestive fate. This study focused on Citri Reticulatae Pericarpium (CRP), the sun-dried pericarp derived from *Citrus reticulata* Blanco and its cultivars [1]. CRP is consumed as a food, a condiment, and a popular dietary supplement, and is officially listed in the Chinese Pharmacopoeia and the United States Pharmacopeia [1,2,3,4]. In fact, CRP is widely used as a drug in traditional Chinese medicine (TCM) with indications to treat nausea, vomiting, indigestion, anepithymia, diarrhea, cough, and expectoration [1]. Among various cultivars of CRP, the pericarp of *Citrus reticulata* ‘Chachi’ (GCP, “Guang Chenpi” in Chinese) is considered to have the highest quality [1,3]. Overall, CRP contains numerous chemical moieties, like alkaloids, flavonoids, and essential oils, among which flavonoids are the primary bioactive constituents [1]. Traditionally, CRP products are aged from one to more than ten years, and TCM practices hold that aging increases the quality of CRP which is scientifically linked to changes in the content of flavonoids and volatile oils [1].

However, long-term storage and aging of CRP may be prone to mold growth or commercial falsification to ramp product costs [1,5]. It was found that storage at high temperature and high relative humidity (RH) for more than 7 days led to the growth of both mold and bacteria on CRP [6], addressing the importance of the optimal condition for storage. Moreover, developing a simple, rapid, and accurate method to classify and characterize CRP products with different aging periods is necessary. To this end, volatile organic compounds (VOCs) could be robust indicators for quality and safety assessments [7]. In fact, VOCs emitted from GCP and other varieties of CRP have been well studied [1,3,4,8,9,10,11,12,13], but there is scant information on VOCs’ emission during the shelf life. Moreover, there exists limited evidence of the precise links between the aging period and chemical compositions of CRP, the content of bioactive moieties, their bioaccessibility and their impact on consumer health. Such gaps could be bridged by using in vitro digestion models to evaluate the digestibility of food and oral formulations or to tackle inter-human variability of differences in digestion between males and females [14,15,16]. Such methods have been successfully applied to study the bioaccessibility of bioactive compounds in citrus peels [17,18,19]. Specifically, flavonoids in citrus peel extracts were sensitive to pH conditions during in vitro gastrointestinal (GI) digestion and showed poor stability and bioavailability [17]. Altogether, there is a growing body of evidence of the differential digestion of foods and oral formulations. However, we could not locate a study on storage conditions, aging time, and the differential digestive performances of CRP in healthy males and females in the scientific literature.

This study aimed to investigate the shelf life of differently aged GCP products and the subsequent digestive fate of the corresponding decoctions. Respiratory emissions (carbon dioxide (CO_2_), oxygen (O_2_), and methane (CH_4_)) were monitored as safety indicators, while key flavonoids (hesperidin, didymin, nobiletin, tangeretin, and 3,5,6,7,8,3′,4′-heptamethoxyflavone (HMF)) were monitored as quality indicators. Specifically, VOCs were explored by proton transfer reaction-time of flight-mass spectrometer (PTR-TOF-MS) as a fingerprinting technique to characterize different GCP products during their aging processes, providing a new, rapid, non-invasive, and efficient quality control method. In turn, we hypothesized that differences in GCP’s fingerprints are closely correlated to its digestive fate and hesperidin release in the guts of both males and females.

## 2. Materials and Methods

### 2.1. Materials

According to the standard outlined in the Product of Geographical Indication—Xinhui Chenpi (DB4407/T 70-2021), GCP products from the Xinhui district must be aged locally for at least three years. Previous studies have shown that the content of bioactive compounds increased in early storage stages, then decreased and stabilized [20], thus, this study focuses on the primary period following three years of aging. GCP aged since 2015, 2016, and 2018 (abbreviated as samples Y2015, Y2016, and Y2018) were purchased from Jiangmen Ligong International Food Co., Ltd. (Jiangmen, China). *Citrus reticulata* ‘Chachi’ was grown and harvested in the Tianma area (Xinhui District, Guangdong Province, China). Once received, GCP pieces were kept in sealed jars and the powders were prepared by pulverization using a small-sized mill (AQ-180E, Nail, Ningbo, China) right before the experiments. These powders were used to prepare GCP decoctions made from samples Y2015 and Y2018 (abbreviated as Y2015-D and Y2018-D). For this, 4 g of fine powders (passing a 60-mesh sieve) were mixed (1:15 *w*/*v*) with 60 mL deionized water for 30 min (90 °C, stirring at 600 rpm). The suspension was filtered through a 100-mesh sieve and the filtrate was cooled down on ice for 10 min before being used in the experiments. 

Methanol, acetonitrile, and formic acid, all in HPLC analytical grade, were obtained from various providers (ThermoFisher, Shanghai, China; Macklin, Shanghai, China; Bio-Lab Ltd., Jerusalem, Israel; Fisher Scientific, Leicestershire, UK). Reference standards (purity ≥98%) of didymin, nobiletin, tangeretin, and 3,5,6,7,8,3′,4′-heptamethoxyflavone (HMF) were purchased from Chengdu RefMedic Biotech Co., Ltd. (Chengdu, China). Reference standards of hesperidin (at least ≥97%) were obtained from two providers (Sigma-Aldrich, St. Louis, MO, USA or Chengdu RefMedic Biotech Co., Ltd., Chengdu, China). For HPLC analyses, deionized water was prepared by using a Milli-Q^®^ IQ-7000 (Merck, Lyon, France) water purification system or purchased from Bio-Lab Ltd. (Jerusalem, Israel). Pepsin from porcine gastric mucosa (P7000, ≥250 units per mg solid), pancreatin from porcine pancreas (P7545), phenylmethanesulfonyl fluoride (PMSF) solution, taurocholic acid sodium salt hydrate (≥95%), magnesium chloride hexahydrate, ammonium carbonate, potassium chloride, potassium dihydrogen phosphate, and pepstatin-A were purchased from Sigma-Aldrich (Rehovot, Israel). Glycodeoxycholate acid sodium salt (≥97%, Holland Moran, Israel) and calcium chloride dehydrate (Spectrum Chemical Manufacturing Corporation, New Brunswick, NJ, USA) were obtained. Simulated gastric fluid (SGF) and simulated duodenal fluid (SDF) were made in distilled water from stock solutions based on bio-relevant information (Table 1).

### 2.2. Proximate Analysis of GCP

The workflow diagram of experimental design is presented in Figure 1. GCP samples were subjected to moisture, ash, volatile solids (VS), lipid, total sugar, crude protein, and multiple elements determinations. Samples were oven-dried (105 °C) and then incinerated (600 °C, 6 h, Muffle Furnace, STM-12-12, Henan Sante Furnace Technology Co., Ltd., Luoyang, China) to determine moisture, ash, and VS contents following standard method No. 2540 [21]. Lipid content was measured by the Mojonnier method [22]. The contents of crude protein, total sugar, and multiple elements including nitrogen (N), phosphorus (P), carbon (C), boron (B), chromium (Cr), cobalt (Co), nickel (Ni), copper (Cu), manganese (Mn), arsenic (As), selenium (Se), strontium (Sr), molybdenum (Mo), cadmium (Cd), tin (Sn), barium (Ba), zinc (Zn), lead (Pb), potassium (K), magnesium (Mg), iron (Fe), sodium (Na), calcium (Ca), aluminum (Al), vanadium (V), and thallium (Tl) were determined by the Kjeldahl method, phenol-sulfuric acid colorimetry method, molybdenum-antimony anti-colorimetric method, potassium dichromate oxidation-external heating method, and ICP-OES/MS method, based on the State Standard of the People’s Republic of China (NY/T 2017-2011, GB/T 15672-2009, GB 5009.5-2016, GB 5009.268-2016) by the Convinced-Test Technology Co., Ltd. (Nanjing, China). All these experiments were carried out at least in duplicates.

### 2.3. Monitoring of GCP During Storage and Accelerated Spoilage

#### 2.3.1. Gas Production During Shelf Life

Researchers have found that the proper storage conditions for CRP to prevent mildew are a temperature below 25 °C and humidity below 85% [1]. It has been reported that after storage at high temperature and high relative humidity for more than 7 days, both mold and bacteria grew on CRP [6]. Thus, two storage scenarios were established: one at 25 °C for two months and another of accelerated spoilage at 35 °C with a similar timeframe, and humidity was closely controlled using a closed circulatory system. Samples of 4 g GCP powders passing through the 24-mesh sieve were placed into a 250 mL glass jar in which a 25 mL vial containing 20 mL of deionized water was loaded, i.e., powders did not come into direct contact with the water. Additionally, an automatic temperature and humidity recorder with an epitaxial probe (KE-COS-03, Kong Sai En) was placed in an empty jar for humidity monitoring of the entire process. Then, all jars were incubated (Jeio Tech Lab Companion incubators) at set temperatures (25 °C or 35 °C) and connected to gas collectors with a two-way valve to create the required aerobic conditions. Gas collectors for each bioreactor were connected to a MicroOxymax respirometry system (Columbus Instruments, Columbus, OH, USA) equipped with a Paramagnetic O_2_ Sensor (Serial 200135-3), a Non-Dispersive Infrared CO_2_ sensor with a 0–3% detection range (Serial 200135-4), a CH_4_ sensor with a 0–5% detection range (Serial 200135-5), and a gas sample drier (Serial 200135-2) [23,24]. Auto-sampling and auto-measurement of the accumulation and the composition of gases were operated by a system sampling pump (Serial 200135-1) every 4 h for each bioreactor. The measured RH in the bioreactor slightly fluctuated due to each auto-sampling of gases, which was 85.2~90.9% at 25 °C and 90.3~99.9% at 35 °C. All these experiments were carried out in triplicates.

#### 2.3.2. Emission of VOCs During Storage and Accelerated Spoilage

Simulated storage (25 °C and 60% RH) and accelerated spoilage (35 °C and 90% RH) were performed for a month for samples Y2015, Y2016, and Y2018. Each GCP piece was cut into a 1 g square sample, and then placed in a 250 mL glass jar and incubated in a constant climate chamber (Memmert HPP110, Memmert GmbH + Co. KG, Schwabach, Germany) with the set conditions. The VOCs’ emission from the samples was detected by the proton transfer reaction-time of flight-mass spectrometer (PTR-ToF-MS 1000, Ionicon Analytik Ges.m.b.H, Innsbruck, Austria) once a day. Conditions of the analyses were the following [23,24]: drift temperature of 80 °C, drift pressure of 2.30 mbar, drift voltage of 630 V, and ratio of electric field strength to number density (E/N) of 142 Td. H_3_O^+^ was the reagent ion, and the mass-scale was calibrated using the signal at the mass to charge ratios (*m*/*z*) of 21.022 and 59.049. The *m*/*z* of the protonated components and the intensities were recorded. The recorded time for each channel was ~120 s with one spectrum per second, and intensities at a stable level were collected (usually from 90 to 120 s) for data analysis. Experiments were carried out in triplicates.

#### 2.3.3. Changes in Flavonoid Levels During Storage and Accelerated Spoilage

Simulations of storage (25 °C and 60% RH) and accelerated spoilage (35 °C and 90% RH) were performed for a month for samples Y2015, Y2016, and Y2018. Samples of 6 g GCP powders passing through the 24-mesh sieve were placed in a 250 mL glass jar and incubated in a constant climate chamber with the above-mentioned conditions (Memmert HPP110, Memmert GmbH + Co. KG, Schwabach, Germany). Every six days, 0.3 g powders were taken and mixed (1:50 *w*/*v*) with 15 mL methanol, and then the solution was sonicated in an ultrasonic bath (50 W, KB-50B, Kunshan Ultrasonic Instruments Co., Ltd.) for 60 min. The supernatant was filtered through a 0.2-μm needle membrane filter into an auto-sampler vial for HPLC analysis by using high-performance liquid chromatography (Agilent Technologies, INC 1220 Infinity, Santa Clara, CA, USA). Chromatographic separation was performed on an Agilent InfinityLab Poroshell 120 SB-C18 (4.6 × 250 mm, 4 µm) with an Agilent InfinityLab Poroshell 120 SB-C18 (4.6 × 5 mm, 4 µm). The method was modified from the one developed by Zeng’s group [25]. The flow rate was 0.6 mL/min, the column temperature was 25 °C and the maximal pressure was 250 bar. The mobile phase was 0.2% formic-water (A) and acetonitrile (B), followed by a gradient elution: 0–6 min, 16–19% B; 6–8 min, 19–44% B; 8–45 min, 44% B. Sample injection volume was 2 µL. Hesperidin and didymin were measured at 283 nm; nobiletin, HMF, and tangeretin were measured at 330 nm. Experiments were carried out in triplicates. To establish the method’s reliability, the calibration curves of the compounds are displayed in Appendix A.

### 2.4. Evaluation of In Vitro Digestive Performance of GCP

#### 2.4.1. Sex-Based In Vitro Gastrointestinal Digestion of GCP Decoction

GCP is typically consumed as a decoction, hence, GCP decoctions prepared from samples Y2015 and Y2018 (abbreviated as Y2015-D and Y2018-D) were subjected to in vitro digestions. Simulated GI digestions were performed in a water-jacketed bioreactor kept at 37 °C and controlled by a dual auto titration unit (Titrando 902, Metrohm, Herisau, Switzerland) that gradually varied the reactor’s pH to simulate gastric and intestinal pH changes. These gradients recreated conditions of healthy adult males or females pre-programmed into the control software (“TIAMO 2.0” software, Metrohm, Herisau, Switzerland) [15,26,27]. Briefly, 40 mL GCP decoction was mixed (2:3 *v*/*v*) with 60 mL simulated gastric fluid (SGF) and CaCl_2_(H_2_O)_2_ (final concentration of 0.15 mM in SGF), pepsin (2000 U/mL or 1600 U/mL in SGF for males or females, respectively) in the bioreactor. The pH was rapidly adjusted using 1 M NaOH to the initial values of 4.5 or 5 for males or females, respectively. Then a computer-controlled gastric pH gradient was initialized by auto titration of 0.3 M HCl to simulate a 2-h gastric phase for males or a 3-h gastric phase for females. In turn, a 2-h intestinal phase was initiated by rapid pH elevation to 6.25 using 0.3 M NaOH, followed by mixing 50 mL gastric chyme (1:1 *v*/*v*) with 50 mL simulated duodenal fluids (SDF) and the addition of pancreatin (100 U/mL), bile salts of sodium glycodeoxycholate (5 mM), CaCl_2_(H_2_O)_2_ (final concentration of 0.6 mM in SDF) and taurocholic acid (5 mM for males or 10 mM for females). The pH was maintained throughout the intestinal phase at 6.25 using 0.3 M NaOH. Aliquots of 1.1 mL samples were aspirated at 0, 30, 60, 120, and 180 min of gastric phase (abbreviated as G0, G30, G60, G120, and G180, G180 only for females), and similarly, duodenal effluents were collected at 30 and 120 min of the intestinal phase (abbreviated as D30 and D120). The samples at time point G0 did not contain digestive enzymes. Gastric and intestinal effluents were inactivated immediately with 25 µL pepstatin A and 25 µL PMSF, respectively, then stored at −20 °C until further analysis. Simulated digestion experiments were carried out in duplicates.

#### 2.4.2. Determination of Bioaccessibility of Hesperidin in Digestive Samples

Each digestive sample was centrifuged for 15 min (stirring at 10,000 rpm, ambient temperature), and then 0.5 mL of the supernatant was mixed (1:5 *v*/*v*) with 2 mL methanol. The solution was sonicated in an ultrasonic bath (35 W, Elmasonic S15, Elma, Singen, Germany) for 30 min, and the supernatant was filtered through a 0.22-μm needle membrane filter to an auto-sampler vial for HPLC analysis with a high-performance liquid chromatography (Agilent 1100 Series). Chromatographic separation was performed on an Agilent ZORBAX Eclipse Plus C18 (4.6 × 250 mm, 5 µm). The content of hesperidin was measured as mentioned previously (Section 2.3.3) with some modifications. Here, the maximal pressure was 350 bar, and the gradient elution was as follows: 0–6 min, 16–19% B; 6–20 min, 19–44% B; 20–25 min, 44% B. Hesperidin was measured at 283 nm. Detailed information regarding the calibration curve is displayed in Appendix A.

The indexes of recovery and bioaccessibility of hesperidin during in vitro GI digestion were calculated according to Equations (1) and (2), respectively [14,28]:recovery index (%) = A/C × 100(1)
bioaccessibility index (%) = B/C × 100(2)

A is the hesperidin content quantified at each time point, B is the hesperidin content quantified in the supernatant after the complete digestion, and C is the hesperidin content quantified in the tested food before digestion (all units were mg/g _dry matter_)_._ The recovery index is related to the percentage of hesperidin in the digest at each time point of the digestion. The bioaccessibility index is defined as the percentage of hesperidin that is solubilized in the chyme after the intestinal phase, thereby defining the proportion of the compounds that could be absorbed into the systematic circulation [14,28].

### 2.5. Experimental Design and Data Analysis

The VOCs’ emission data were processed and analysed by PTR-MS Viewer 3.4 (Ionicon Analytik Ges.m.b.H., Innsbruck, Austria). Principal component analysis (PCA) was carried out using the R programming language (Version 1.4.1717—© 2009–2017 RStudio) to analyze the accumulative VOC emitted by GCP under different storage conditions over a month. The three-dimensional (3D) plots of the spectra profiles of VOCs were created using MATLAB R2021b (The MathWorks, Inc., Natick, MA, USA). The tentative structures of VOCs were drawn using ChemDraw (PerkinElmer Informatics 22.0.0). Orthogonal partial least squares discriminant analysis (OPLS-DA) was obtained and the corresponding variable importance in projection (VIP) values were calculated using SIMCA 14.1 (Umetrics, Umeå, Sweden). Heatmaps were generated using average normalized raw data with log(10) transformation to visualize the intensity of VOCs emitted at various storage timepoints. Statistical analyses for analysis of variance (ANOVA) tests were conducted, and the data were plotted by GraphPad Prism 9.2.0 (San Diego, CA, USA), where the significance was denoted as the letter where *p* < 0.05. All reported data represent the mean ± standard deviation of at least duplicate experiments.

## 3. Results and Discussion

### 3.1. Proximate Analysis of GCP 

To evaluate the quality of aged GCP, contents of moisture, VS, ash, lipid, crude protein, total sugar, and multiple elements were determined for GCP samples and are shown in Table 2. The aging periods of GCP showed a significant effect on the fat content, total sugar content and the contents of the elements C, Sn, Pb, Fe, Al, and V (*p* < 0.05). These analyses showed all samples had marginal contents of heavy metals like Pb, Cd, and Tl. Our results are consistent with previous studies showing that CRP contains various trace elements such as K, Na, Ca, Mg, Cu, Zn, Fe, Sr, Mn, Mo, and Se [1]. Additionally, the findings indicate that the aging process was accompanied by a significant rise of Sn and Fe levels as well as a reduction in the total sugar content (*p* < 0.05). This drop in sugar content concurs with previous results and is hypothesized to arise from thermal degradation processes due to the prolonged aging [29]. Therefore, the total sugar content can be used as one of the indexes of the aging period.

### 3.2. Monitoring of GCP During Storage and Accelerated Spoilage

#### 3.2.1. Gas Production During Shelf Life

Aging and spoilage of GCP are related to respiration, growth of microorganisms including yeast and molds, making O_2_, CO_2_, and CH_4_ clear indicators [30,31,32]. This work examined two real-life storage scenarios over two months and to our knowledge, this is the first study to monitor such indicators of GCP with results given in Figure 2. Both aging periods of GCP and the storage conditions significantly affected the lag phase duration (when the initial O_2_ consumption rates or the CO_2_ accumulative rates were <1 μg/min) (*p* < 0.05); such durations were in an ascending order among samples: Y2016 < Y2018 < Y2015 without correlation to their aging periods and were accompanied by a decrease at higher temperature and humidity in accelerated spoilage (Figure 2D,E). The lag phase might indicate the rate at which microbial community establish itself on the substrate, providing deeper insights into its nutritional richness or microbial toxicity. After the lag phase, the signal of high microbial activity for all samples appeared manifesting as the sharp rises of O_2_ consumption rates and CO_2_ accumulative rates (Figure 2D,E), along with a dramatical increase in the O_2_ consumptions and the CO_2_ accumulations (Figure 2A,B). Such a high microbial activity might be an early signal of spoilage. However, the unexpected earlier signal of spoilage (12~17 days) for samples at the storage conditions (25 °C) could be explained by the relatively high RH (85.2~90.9%) caused by the evaporation of water in the vial loaded in the jar. Similar to this study, the CO_2_ respiration rate or CO_2_ formation increased with increasing temperature for wheat [32], shredded cabbage [30], and skinless chicken breast [33].

Subsequently, the CO_2_ accumulative rates and the O_2_ consumption rates of all samples under both storage conditions displayed gradual decreasing trends until the spoilage was almost complete at about 60 days (when CO_2_ accumulative rates were <6 μg/min or O_2_ consumption rates were <5 μg/min) (Figure 2D,E); the corresponding O_2_ consumptions and the CO_2_ accumulations increased slowly until becoming relatively stable and were eventually in ascending order among the samples: Y2015 < Y2016 < Y2018 under either condition (Figure 2A,B). A significant rise in total sugar content along the longer aging time of GCP (*p* < 0.05) suggested that the oxidation of sugar in a respiratory reaction could contribute to the total CO_2_ emission (Table 2). Additionally, the degree of spoilage was related to the increase of the CO_2_ amount, and higher storage temperature contributed to the increase of CO_2_ formation [33]. Therefore, raising the temperature and the humidity increased the spoilage levels of GCP as demonstrated in the higher contents of CO_2_ accumulations and O_2_ consumptions eventually (Figure 2A,B). Distinctively, the CH_4_ accumulative rates of samples decreased primally and then fluctuated limitedly, eventually resulting in little negative CH_4_ accumulations (Figure 2C,F). Though food spoilage can be tested by detecting naturally emitted gases like CH_4_ as food decays [34], CH_4_ cannot serve as an indicator of the aerobic spoilage of GCP. Overall, the respiration of GCP might be monitored with O_2_ and CO_2_ and thus they were validated as indicators of the early signals of spoilage.

#### 3.2.2. Emission of VOCs During Storage and Accelerated Spoilage

By constantly monitoring the VOCs’ emission from samples Y2015, Y2016, and Y2018 during simulations of storage (25 °C and 60% RH) and accelerated spoilage (35 °C and 90% RH) for one month, plots of the VOCs’ emission were created (Appendix A), where four compounds (with *m*/*z* of 32, 81, 95, and 137) with the relatively highest intensities were found in all samples under both simulations. These compounds (compounds A1~A4) were proposed to serve as the characteristic VOCs of GCP and are presented in Table 3, where the chemical compositions and the tentative identifications taking into account the available volatile compounds of GCP are suggested as well [1,3,9,10,11,12,13]. Methylpyrazine (tentatively A3) could serve as a marker to discriminate between GCP and the dried peel of other *C. reticulata* Blanco cultivars (CP, “Chenpi” in Chinese) with higher content in GCP than CP [3]. Additionally, monoterpene (C_10_H_16_) with various isomers found in GCP (tentatively A4) were usually the main components of the volatiles of sun-dried citrus peel, and that they could be considered as the characteristic compounds of GCP and the markers to discriminate between GCP and CP [3,9]. Furthermore, a score plot of the PCA for spectra profiles was carried out to identify a volatile fingerprint of GCP and the storage conditions, which showed a relatively similar pattern for three samples but a significant difference between the two simulated conditions (Figure 3A). Thus, VOCs’ emission from GCP varied with temperature and humidity. 

Additionally, the PCA plot for VOCs’ emission during the first week of simulated storage manifested the VOCs differences among the samples, where Y2015 and Y2018 exhibited unique emission patterns separately, but both had a similarity to Y2016 (Figure 3B). Then, the supervised OPLS-DA was used to further identify the VOCs variation among samples (Figure 3C). Statistical parameters (R^2^X = 0.981, R^2^Y = 0.929, Q^2^ = 0.799) of the OPLS-DA model showed a good fit and prediction, and the model was validated using 200 permutation tests (Appendix A). By using the one-way ANOVA test, compounds with VIP values > 1.00 and *p* < 0.05 (compounds B1~B18) were selected as the potential markers to discriminate among the aging periods of GCP and are listed according to the VIP values in Table 4, together with the suggested tentative identifications [1,3,4,8,9,10,11,12,13]. Furthermore, a heatmap was applied to visualize the variations in these compounds, where colors indicate the signal intensity (log10-transformed) of each metabolite (Figure 4A). The majority of compounds showed higher levels in Y2015 than in Y2016 and Y2018 with a little difference between Y2016 and Y2018 (Figure 4A), consistent with the PCA results shown in Figure 3B. Thus, the aging periods of GCP contributed to VOCs accumulation and subsequent emission, which was consistent with the findings that storage time affected the volatile oil content of CRP [1]. Aging at least one year longer than samples Y2016 and Y2018 resulted in an increasing VOCs emission in Y2015, thus a turning point is essential for the quality change in GCP during aging. Similarly, the levels of metabolites that contributed significantly to identifying the storage years of GCP showed a rising trend during 3–10 years and peaked at 5–10 years [34]. Furthermore, methyl acetate (tentatively B1) may cause aromas of ripe fruits such as tangerines, apples, and red berries in GCP [13], and ethyl acetate (tentatively B7) was used as a quality index of citrus aroma [9]. Additionally, compounds (3Z)-hex-3-en-1-ol, 2-hexanone, 2-hexen-1-ol, and hexanal (tentatively B4), compound butanone (tentatively B5), compounds 3-methylbutanal and 2,3-butanedione (tentatively B9), compound methylpyrazine (tentatively B15), compounds thymol and perilla aldehyde (tentatively B11), compounds (R)-(+)-β-citronellol and decanal (tentatively B17) could be considered as the discriminating markers of GCP and CP [3,4,9]. Specially, compounds 5-isopropyl-2-methylphenol, 2-methoxy-4-vinylphenol, and thymol (tentatively B11) could be key aroma compounds of aged GCP in different years [8].

The OPLS-DA with statistical parameters (R^2^X = 0.908, R^2^Y = 0.974, Q^2^ = 0.961) showed a clear separation in VOCs variation between the first week and the fourth week of simulated spoilage (Figure 3D) and was validated using 200 permutation tests (Appendix A). By using the one-way ANOVA test, compounds with VIP values > 1.10 and *p* < 0.05 (compounds C1~C8) were selected as potential discriminates of the spoilage levels and are listed in Table 5, together with the suggested tentative identifications [1,3,4,8,9,10,11,12,13]. Furthermore, a heatmap visualizing the variations in these compounds showed that all compounds showed an upward trend in the fourth week compared to the first week of accelerated spoilage (Figure 4B), thus they may be related to the spoilage level of GCP manifesting as the higher intensity at the higher spoilage level. Overall, temperature and humidity significantly affected the VOCs emission of GCP, among which compounds B1~B18 showed the potential to descriptively measure the aging process and compounds C1~C8 showed the potential to discriminate among the spoilage levels.

#### 3.2.3. Changes in Flavonoid Levels During Storage and Accelerated Spoilage

The flavonoids of CRP have various important pharmaceutical activities [1,10]. The contents of five key flavonoids (hesperidin, didymin, nobiletin, tangeretin, and HMF) during the simulated storage (25 °C, 60% RH) and the accelerated spoilage (35 °C, 90% RH) for one month are shown in Figure 5. Both the duration of storage simulation and the aging periods of GCP significantly affected the contents of five flavonoids under either simulation (*p* < 0.05) (Figure 5). Previous studies have similarly demonstrated that contents of flavonoids are influenced by aging duration [20]. The contents of hesperidin and didymin in all samples displayed relatively decreasing trends under either storage or accelerated spoilage, and those contents in each sample were relatively higher at the end of the simulated storage than that of accelerated spoilage, whereas the contents of the other flavonoids in each sample were similar at the end of either process (Figure 5). Thus, improper storage conditions might lower the quality of GCP by reducing the contents of hesperidin and didymin. Significant differences in the contents of nobiletin, tangeretin or HMF existed initially among the three samples and were sustained until the end of both the simulated storage and the accelerated spoilage (*p* < 0.05) (Figure 5C–E). Noticeably, the contents of nobiletin and tangeretin significantly increased with the aging time of GCP and those contents in sample Y2018 were significantly the lowest (*p* < 0.05) (Figure 5C,E). In agreement with the previous study [1], the quality of GCP was increased with the extension of the aging time. It was found that nobiletin, tangeretin, and HMF increased with the storage period of CRP [1,35,36], the insignificant correlation between the contents of hesperidin and HMF and the aging periods of the samples was likely due to the insufficient difference among the aging periods. Our findings also raised another question about the possible links between the aging periods of GCP and the digestibility in males and females as well as consumer health.

### 3.3. Evaluation of In Vitro Digestive Performance of GCP

Hesperidin is a major phenolic compound found in citrus peels [17], which was chosen as a model for the flavonoids in this study. The hesperidin contents in the decoction samples of Y2015-D and Y2018-D under in vitro digestive conditions of healthy males and females are shown in Figure 6A. The hesperidin content of two samples decreased significantly at 30 min of gastric phase (G30) under either digestive condition, then remained without a significant change during the gastric and the intestinal phases (*p* < 0.05) (Figure 6A). The corresponding recovery index (%) of hesperidin is depicted in Figure 6B, both the digestion time and the aging periods of GCP showed significant effects on the changes in hesperidin recovery, but the consumer’s sex did not (*p* < 0.05). The recovery of hesperidin for samples at both digestive conditions reduced to ~30% or ~40% at the gastric phase, mainly at G30, and subsequently remained without a significant difference until the end of digestion (Figure 6B). Similarly, a significant decrease of hesperidin was found in the sweet orange pulp during the gastric phase and did not differ between the gastric and intestinal phases [37], and hesperidin in the pulp of Satsuma, Ponkan, and Naval orange was significantly decreased after the in vitro digestion [38]. Additionally, a previous study investigating the in vitro digestion of the dried tangerine peel has shown that the total flavonoid content (TFC) was reduced greatly by the SGF treatment, but was not affected by the SIF treatment [19]. The bioactivity of phenolic compounds including flavonoids, is related to phenolic-protein interactions [19], thus; the efficient release of hesperidin from the matrix in the gastric or intestinal phase and the binding reaction with pepsin or pancreatin might lead to a reduction in recovery. Moreover, Sun et al. suggest that a decrease in hesperidin after simulated in vitro digestion could be partially explained by its transformation into chalcones under alkaline conditions, or by racemization creating enantiomers due to the pH changes, which made them escape HPLC detection [38].

The biochemical conditions in the intestinal phase caused the increasing trends in the content and recovery of hesperidin, except for sample Y2018-D under the digestive condition of females, although these trends were not significant (*p* < 0.05) (Figure 6A,B). Such trends may be caused by the transformation from other flavonoids or be related to the additional 2-h digestion and the effect of the intestinal enzymes and bile salts, which helped the release of phenolics bound to the matrix [14,38]. Moreover, the bioaccessibility of hesperidin of samples under both digestive conditions ranged between ~30% and ~50%, and the consumer’s sex and aging periods of GCP did not show a significant effect on it (Figure 6C). Overall, the differences in GCP’s fingerprints were closely associated with its digestive fate and hesperidin release in the guts of healthy males and females without a significant impact on the bioaccessibility of hesperidin. The obtained knowledge may provide a basis for the possible control of the sex-based digestive performance of GCP.

## 4. Conclusions

The shelf life of traditional Chinese medicine, like GCP, may vary over time and with diverse storage conditions, potentially affecting its digestive fate and overall impact on consumers. Therefore, this study explored the characteristics and shelf life of variably aged GCP as well as their breakdown during simulated digestive conditions of healthy males and females. Monitoring respiration of GCP established O_2_ and CO_2_ as early-stage indicators of spoilage that could prospectively be used as safety measures as well. This study showed that temperature and humidity significantly affected the emitted VOCs during storage, with 18 specific moieties identified as possible descriptive measures of aging and 8 specific moieties identified as potential discriminants of spoilage levels. With respect to possible bioactive moieties, hesperidin, didymin, nobiletin, tangeretin, and HMF were determined as quality indicators, with their contents significantly affected by the duration of the storage simulation and the aging periods of GCP. Notably, nobiletin and tangeretin levels were found to increase over aging time while hesperidin and didymin levels were compromised by improper storage conditions. Decoctions of GCP were found to be similarly digested under in vitro digestive conditions of healthy males or females. In fact, the bioaccessibility of hesperidin ranged from ~30% to ~50% and was not significantly affected by the GCP’s aging nor by the consumer’s sex (*p* < 0.05). Thus, this work demonstrates that GCP’s aging and spoilage can be analytically monitored with an omics approach. Interestingly, this work provides evidence that the bioactive hesperidin is likely to be equally bioaccessible in males and females. In the face of the growing interest in nutraceuticals and Chinese medicine, future experiments are needed to validate the proposed markers of quality and spoilage as well as affirm the digestive fate of GCP in different consumers.

## Figures and Tables

**Figure 1 foods-13-03671-f001:**
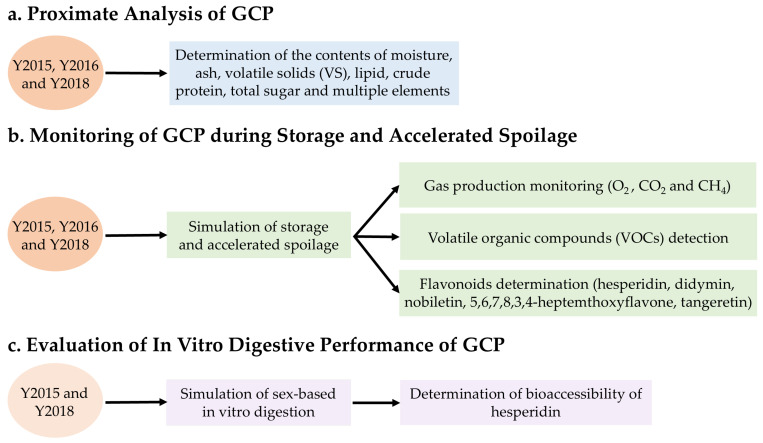
Workflow diagram of experimental design.

**Figure 2 foods-13-03671-f002:**
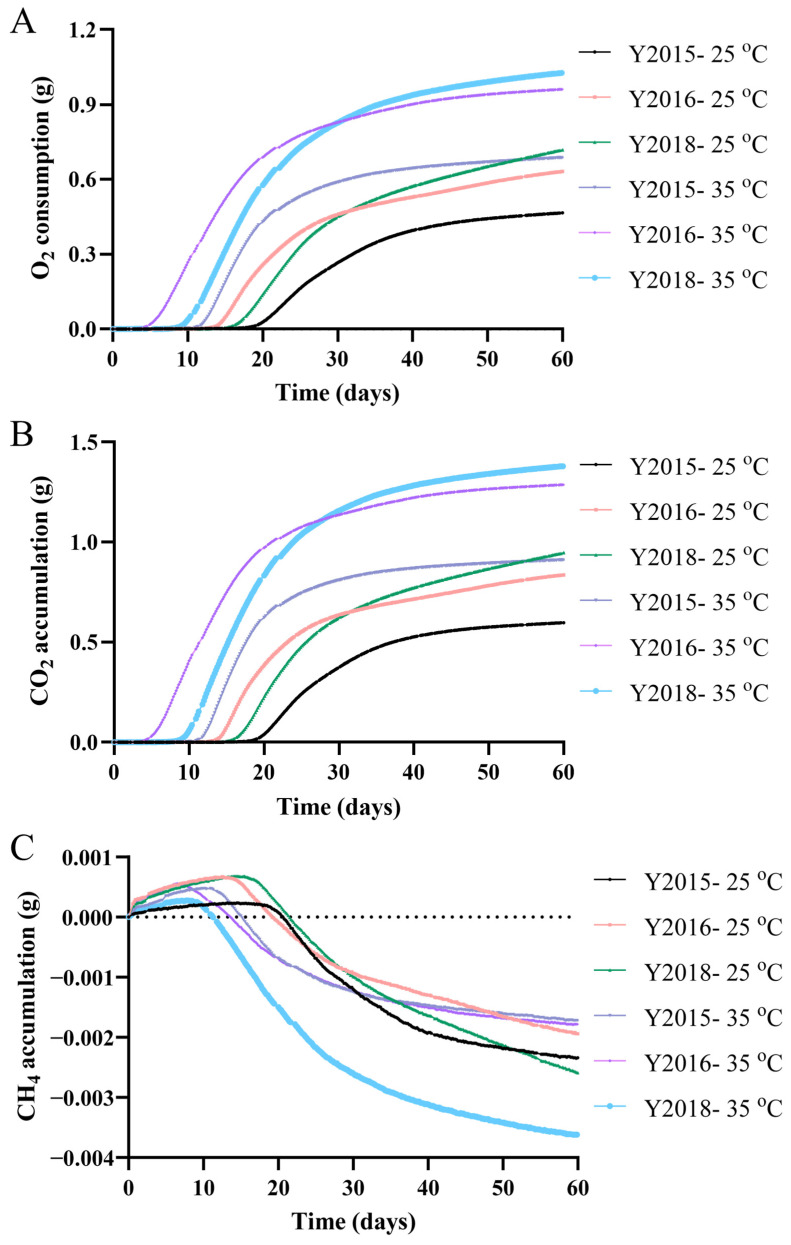
Respiration under simulated storage (25 °C and 85.2~90.9% RH) and accelerated spoilage (35 °C and 90.3~99.9% RH) for Y2015, Y2016, and Y2018 for two months. (**A**) O_2_ consumption, (**B**) CO_2_ accumulation, (**C**) CH_4_ accumulation, (**D**) O_2_ consumption rate and the insert of lag phase duration, (**E**) CO_2_ accumulative rate and the insert of lag phase duration, (**F**) CH_4_ accumulative rate. The lag phases were supposed when the initial O_2_ consumption rates or the CO_2_ accumulative rates were <1 μg/min. The *p*-value was obtained by a two-way analysis of variance (ANOVA) test. Different letters indicated a significant difference between samples (*p* < 0.05). Data represented the means of three replicate bioreactors.

**Figure 3 foods-13-03671-f003:**
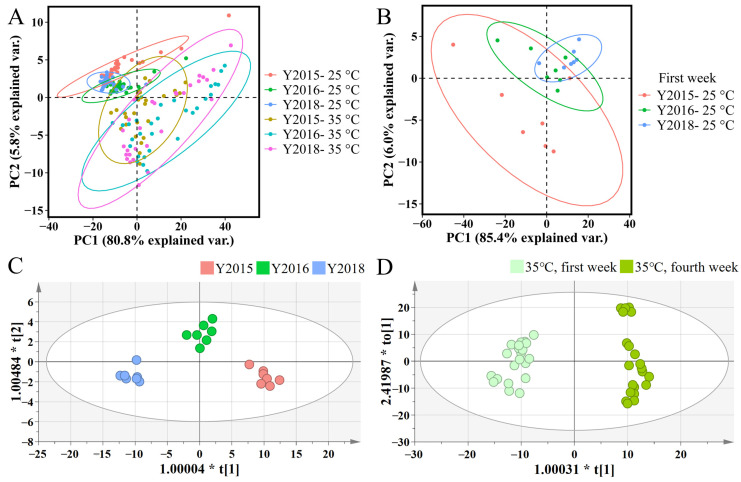
(**A**) PCA plot of the accumulative VOCs emitted by *Citrus reticulata* ‘Chachi’ (GCP) under simulated storage (25 °C and 60% RH) and accelerated spoilage (35 °C and 90% RH) for a month. (**B**) PCA plot of the accumulative VOCs emitted by GCP under simulated storage (25 °C and 60% RH) during the first week of a month. (**C**) OPLS-DA score plots for GCP under simulated storage (25 °C and 60% RH) in the first week (1~7 day) of a month with statistical parameters (R^2^X = 0.981, R^2^Y = 0.929, Q^2^ = 0.799). (**D**) OPLS-DA score plots for GCP under accelerated spoilage (35 °C and 90% RH) in the first (1~7 days) and fourth week (24~30 days) with statistical parameters (R^2^X = 0.908, R^2^Y = 0.974, Q^2^ = 0.961).

**Figure 4 foods-13-03671-f004:**
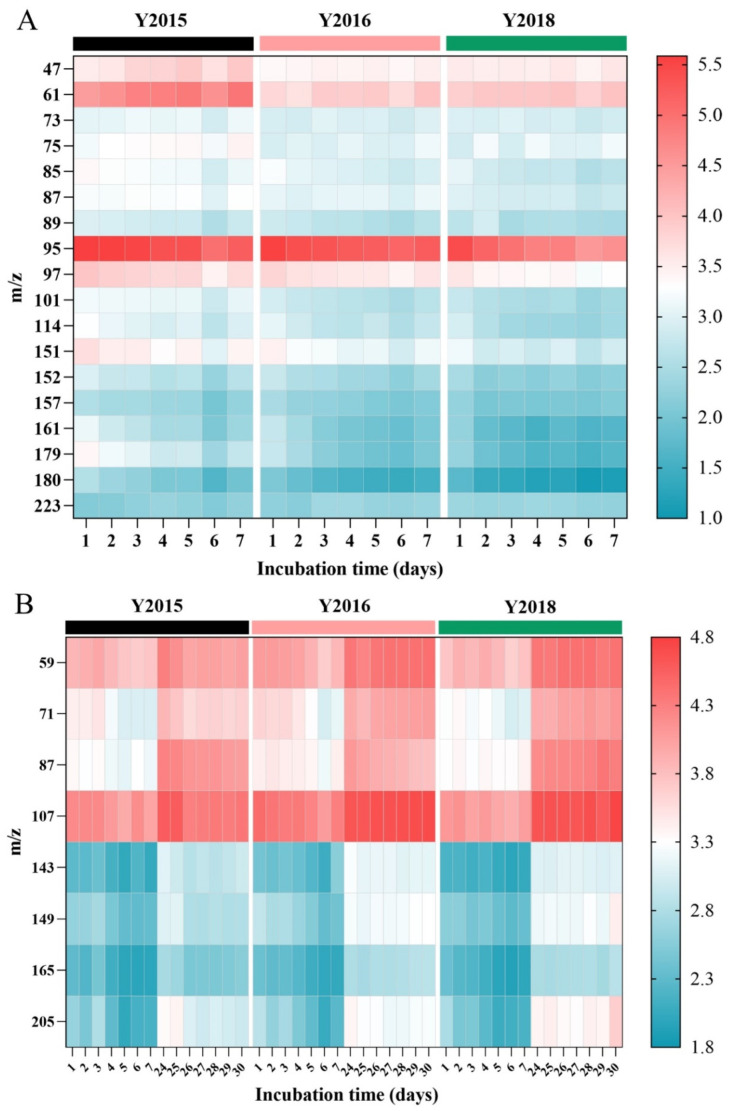
Heat-maps (log10 scales) indicating VOCs intensity from low (blue) to high (red) for different storage timepoints. (**A**) Heat-map for the first week (1~7 days) under simulated storage (25 °C and 60% RH). (**B**) Heat-map for the first week (1~7 days) and the fourth week (24~30 days) under accelerated spoilage (35 °C and 90% RH).

**Figure 5 foods-13-03671-f005:**
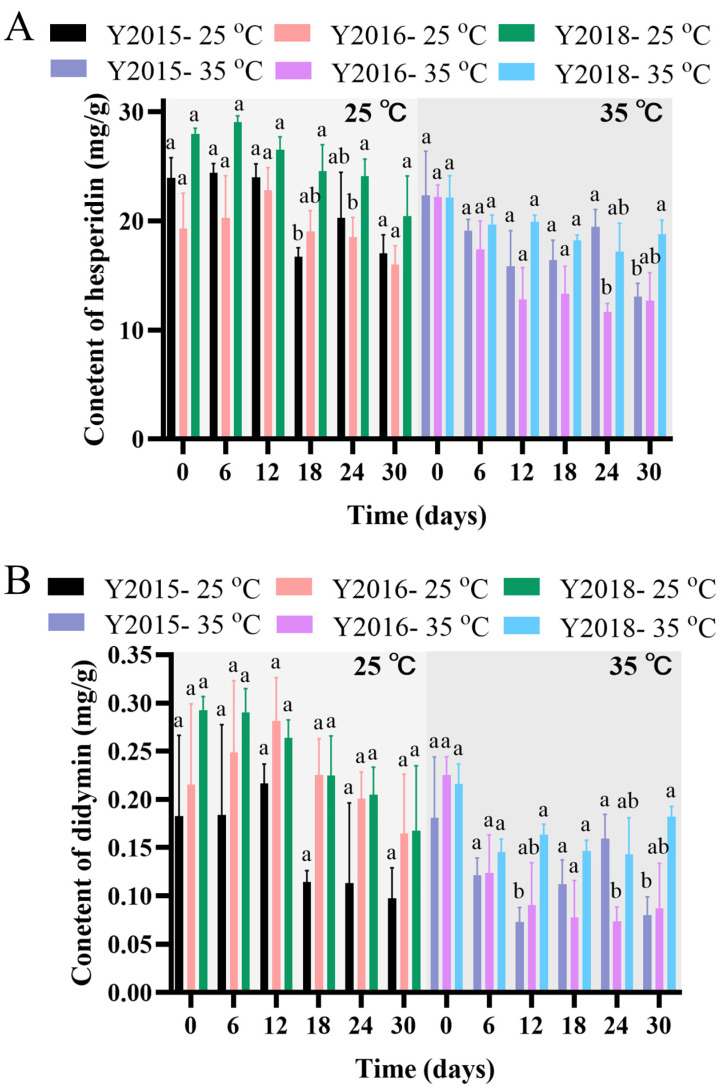
Contents of (**A**) hesperidin, (**B**) didymin, (**C**) nobiletin, (**D**) 3,5,6,7,8,3′,4′-heptamethoxyflavone (HMF), and (**E**) tangeretin in samples Y2015, Y2016 and Y2018 during simulated storage (25 °C and 60% RH) and accelerated spoilage (35 °C and 90% RH) for a month. Data represented the mean of three replicate bioreactors. The *p*-value was obtained by a two-way ANOVA test, different letters indicated a significant difference among three samples at each time section (*p* < 0.05).

**Figure 6 foods-13-03671-f006:**
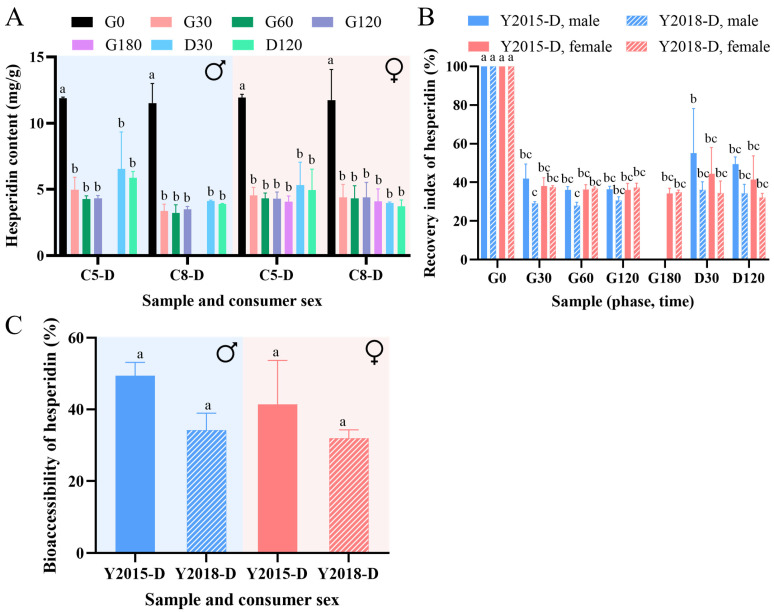
(**A**) Hesperidin contents in Y2015-D and Y2018-D during sex-based in vitro digestion, the *p*-value was obtained by a one-way ANOVA test for each sample, different letters indicated a significant difference among time sections within each group (*p* < 0.05). (**B**) Recovery index of hesperidin during in vitro digestion, the *p*-value was obtained by a three-way ANOVA test (assuming results at G180 for males equal to those at G120), different letters indicated a significant difference among samples (*p* < 0.05). (**C**) The in vitro bioaccessibility of hesperidin, the *p*-value was obtained by a two-way ANOVA test, different letters indicated a significant difference among samples (*p* < 0.05). Data represented the mean of two replicates.

**Table 1 foods-13-03671-t001:** Composition of SGF and SDF for the preparation of 1L solution.

Parameter	Stock (g/L)	SGF (mL)	SDF (mL)
KCl	46.72	11	10.8
KH_2_PO_4_	68	1.8	1.6
NaCl	120	35.2	60.1
MgCl_2_(H_2_O)_6_	30	0.8	2.2
(NH_4_)_2_CO_3_	48	1	-
DW	-	To be completed up to 1L after pH correction	
pH	-	3	7

**Table 2 foods-13-03671-t002:** Proximate analysis of samples Y2015, Y2016 and Y2018.

Parameter	Y2015	Y2016	Y2018
Moisture content (g water/g DS)	0.16 ± 0.01 a	0.11 ± 0.01 a	0.13 ± 0.01 a
Volatile solids content (%DS)	96.17 ± 0.45 a	96.42 ± 0.64 a	96.89 ± 0.90 a
Ash content (g/g DS)	0.038 ± 0.004 a	0.036 ± 0.006 a	0.031 ± 0.009 a
Lipid content (g/100 g)	3.75 ± 0.77 a	3.71 ± 0.38 a	2.64 ± 0.53 b
Crude protein content (g/100 g)	9.91 ± 0.13 a	8.62 ± 0.02 a	9.33 ± 0.16 a
Total sugar content (mg/gDW)	420.66 ± 37.21 b	449.88 ± 14.95 a	451.39 ± 6.39 a
P content (g/kg)	1.10 ± 0.02 a	1.06 ± 0.10 a	0.86 ± 0.16 a
C content (g/kg)	347.09 ± 13.92 b	341.36 ± 12.92 b	441.27 ± 6.44 a
N content (g/kg)	16.95 ± 0.52 a	15.77 ± 0.18 a	15.35 ± 0.14 a
B content (mg/kg)	28.94 ± 2.48 a	25.46 ± 1.94 a	20.58 ± 2.91 a
Cr content (mg/kg)	5.33 ± 0.18 a	5.78 ± 0.27 a	5.28 ± 0.05 a
Co content (mg/kg)	0.08 ± 0.003 a	0.10 ± 0.002 a	0.09 ± 0.0007 a
Ni content (mg/kg)	1.43 ± 0.02 a	1.73 ± 0.02 a	1.64 ± 0.05 a
Cu content (mg/kg)	6.33 ± 0.31 a	5.85 ± 0.39 a	6.19 a
Mn content (mg/kg)	12.65 ± 0.36 a	14.04 ± 2.64 a	15.54 ± 0.27 a
As content (mg/kg)	0.071 ± 0.005 a	0.067 ± 0.003 a	0.065 ± 0.005 a
Se content (mg/kg)	1.56 ± 0.14 a	1.92 ± 0.17 a	1.69 ± 0.05 a
Sr content (mg/kg)	26.31 ± 0.25 a	28.01 ± 1.85 a	22.99 ± 0.53 a
Mo content (mg/kg)	0.11 ± 0.002 a	0.12 ± 0.00008 a	0.11 ± 0.006 a
Cd content (μg/kg)	6.46 ± 0.61 a	15.81 ± 2.12 a	15.54 ± 0.89 a
Sn content (μg/kg)	67.16 ± 1.46 a	61.17 ± 7.21 ab	51.75 ± 5.29 b
Ba content (mg/kg)	22.67 ± 0.11 a	16.32 ± 0.30 a	16.11 ± 0.08 a
Zn content (mg/kg)	10.97 ± 0.17 a	10.39 ± 0.53 a	7.68 ± 0.04 a
Pb content (μg/kg)	253.35 ± 18.23 b	322.72 ± 11.45 a	332.65 ± 2.70 a
K content (g/kg)	10.19 ± 0.03 a	11.27 ± 0.74 a	6.37 ± 0.01 a
Mg content (g/kg)	0.57 ± 0.01 a	0.56 ± 0.02 a	0.56 ± 0.01 a
Fe content (mg/kg)	112.99 ± 1.95 a	104.98 ± 13.98 a	74.64 ± 0.05 b
Na content (g/kg)	0.016 ± 0.003 a	0.031 ± 0.01 a	0.021 ± 0.01 a
Ca content (g/kg)	3.97 ± 0.17 a	3.10 ± 0.14 a	3.30 ± 0.11 a
Al content (mg/kg)	72.69 ± 0.88 b	70.07 ± 1.76 b	116.28 ± 6.77 a
V content (μg/kg)	128.69 ± 1.88 a	130.37 ± 1.25 a	73.11 ± 1.19 b
Tl content (μg/kg)	4.83 ± 0.05 a	8.05 ± 1.91 a	8.79 ± 1.61 a

Data represented the mean of at least two replicates. The *p* value was obtained by the two-way ANOVA test, different letters indicated a significant difference among samples for each testing parameter (*p* < 0.05).

**Table 3 foods-13-03671-t003:** Characteristic VOCs emitted from GCP (with the relatively highest intensities) and detected during simulated storage (25 °C and 60% RH) and accelerated spoilage (35 °C and 90% RH).

No.	*m*/*z*	Protonated ChemicalFormula ^*a*^	Tentative Identification *^b^*	Tentative Structure *^c^*	Refs.
A1	32	(CH_5_N)H^+^ fragment; O_2_^+^ fragment	Not identified; Not identified	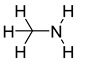	-
A2	81	(C_6_H_8_)H^+^ fragment	1,4-Cyclohexadiene fragment */Bicyclo[3.1.0]hex-2-ene fragment *	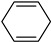	[12]
A3	95	(C_5_H_6_N_2_)H^+^	Methylpyrazine	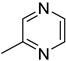	[3]
A4	137	(C_10_H_16_)H^+^; (C_8_H_8_O_2_)H^+^	D-Limonene/α-Pinene/β-Pinene/Limonene/Myrcene/α-Phellandrene/β-Phellandrene/α-Terpinene/γ-Terpinene/Sabinene/Terpinolene/Camphene/13,7-dimethyl-3,6-Octatriene/Ocimene/1-Methyl-4-(1-methylethylidene) cyclohexene/α-Thujene/3-Carene/4-Carene/trans-1,2-Bis(1-methylethenyl)-cyclobutene/4-Methyl-1-(1-methylethyl) bicyclo[3.1.0]hexane didehydro deriv; Benzoic acid methyl ester	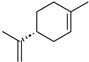	[1,3,9,10,11,13]

*^a^* Chemical formulas were predicted using the PTR-MS Viewer 3.4 software based on a library of recognized materials, that concurs with the literature as well. *^b^* Tentative compounds were suggested according to the chemical formula. Symbol “/” indicated the possible existence of more tentative identifications corresponding to one chemical formula. *^c^* One chemical structure was suggested according to the chemical formula. * The fragment was proposed as a partial chemical compound of the detected volatile component.

**Table 4 foods-13-03671-t004:** Potential VOCs markers that can discriminate between its aging periods (VIP values ≥ 1 and *p* < 0.05) of GCP.

No.	*m*/*z*	VIP	Protonated ChemicalFormula ^*a*^	Tentative Identification *^b^*	Tentative Structure *^c^*	Refs.
B1	75	2.29	(C_3_H_6_O_2_)H^+^; (C_4_H_10_O)H^+^	Hydroxyacetone/Propionic acid/Methyl acetate; 2-Methylpropanol/2-Butanol/n-Butanol	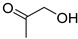	[3,9,13]
B2	47	2.18	(C_2_H_6_O)H^+^	Ethanol	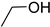	[13]
B3	61	2.15	(C_3_H_8_O)H^+^	1-Propanol	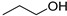	[9]
B4	101	1.69	(C_6_H_12_O)H^+^; (C_5_H_8_O_2_)H^+^	(3Z)-Hex-3-en-1-ol/Hexanal/2-Hexanone/2-Hexenol/2-Hexen-1-ol; 2,3-Pentadione	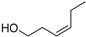	[3,9,13]
B5	73	1.54	(C_4_H_8_O)H^+^	Butanone/Butanal/Tetrahydrofuran	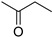	[3,9,13]
B6	179	1.33	(C_14_H_10_)H^+^	Phenanthrene/Anthracene	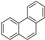	[10]
B7	89	1.30	(C_4_H_8_O_2_)H^+^; (C_5_H_12_O)H^+^	Ethyl acetate/3-Hydroxy-2-butanone/Butanoic acid/1,4-Dioxane; 2-Methyl-1-butanol/3-Methyl-1-butanol	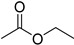	[9,13]
B8	180	1.25	(C_10_H_13_NO_2_)H^+^	Benzoic acid, 2-(methylamino)-, ethyl ester	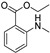	[11]
B9	87	1.23	(C_5_H_10_O)H^+^; (C_4_H_6_O_2_)H^+^	2-Methylbutanal/Pentanal/3-Methylbutanal/1-Penten-3-ol/3-Pentanone/2-Pentanone; 2,3-Butanedione	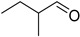	[9,13]
B10	223	1.20	(C_15_H_26_O)H^+^	Patchouli alcohol/Elemol/α-Cadinol/Nerolidol/(−)-Globulol/α-Eudesmol/T-muurolol/Cyclohexanemethanol, 4-ethenyl-a,a,4-tri-methyl-3-(1-methylethenyl)-, [1R-(1a,3a,4a)]-	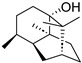	[1,10,11,12]
B11	151	1.14	(C_9_H_10_O_2_)H^+^; (C_10_H_14_O)H^+^	p-Vinylguaiacol/2-Methoxy-4-vinylphenol; p-Cymen-7-ol/2-Methyl-5-(1-methylethyl)-phenol/5-Methyl-2-(1-methylethyl)-phenol/Carvacrol/Perilla aldehyde/Carvone/Thymol/4-(1-Methylethenyl)-1-cyclohexene-1-carboxaldehyde/3-Methyl-4-isopropylphenol/2-(4-Methylphenyl)propan-2-ol/Perillene/D-carvone/Piperitenone/5-isopropyl-2-methylphenol/1-Cyclohexene-1-carboxaldehyde, 4-(1-methyl-ethenyl)-	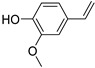	[1,4,8,9,10,11,12]
B12	85	1.07	(C_5_H_8_O)H^+^	(E)-2-Pentenal/3-Methyl-2-butenal/1-Penten-3-one/3-Methyl butynol	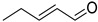	[8,9,13]
B13	152	1.04	(C_8_H_9_NO_2_)H^+^	Methyl 2-aminobenzoate	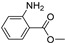	[10]
B14	97	1.03	(C_5_H_4_O_2_)H^+^	Furfural/2-Furan-methyl aldehyde	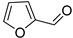	[1,8,9,12,13]
B15	95	1.03	(C_5_H_6_N_2_)H^+^	Methylpyrazine	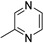	[3]
B16	114	1.03	(C_7_H_14_O)^+^	Heptanal	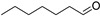	[9,13]
B17	157	1.02	(C_10_H_20_O)H^+^	Decanal/(R)-(+)-β-Citronellol/1-Menthol/Citronellol	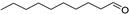	[1,3,4,8,9,10,13]
B18	161	1.02	(C_12_H_16_)H^+^ fragment	Not identified	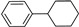	-

*^a^* Chemical formulas were predicted using the PTR-MS Viewer 3.4 software based on a library of recognized materials, that concurs with the literature as well. *^b^* Symbol “/” indicated the possible existence of more tentative identifications corresponding to one chemical formula. *^c^* One chemical structure was suggested according to the chemical formula.

**Table 5 foods-13-03671-t005:** Potential VOCs markers that can discriminate between spoilage levels (VIP values ≥ 1.1 and *p* < 0.05) of GCP.

No.	*m*/*z*	VIP	Protonated ChemicalFormula ^*a*^	Tentative Identification *^b^*	Tentative Structure *^c^*	Refs.
C1	143	1.19	(C_9_H_18_O)H^+^	Nonanal	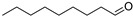	[1,3,8,9,10,12]
C2	165	1.15	(C_10_H_12_O_2_)H^+^; (C_11_H_16_O)H^+^	Phenylacetic acid ethyl ester/Eugenol/Ethyl phenylacetate/Benzene,4-ethenyl-1,2-dimethoxy-; 2-Isopropyl-5-methylanisole/2-Isopropyl-4-methylanisole	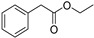	[8,9,10,11,13]
C3	87	1.23	(C_5_H_10_O)H^+^; (C_4_H_6_O_2_)H^+^	2-Methylbutanal/Pentanal/3-Methylbutanal/1-Penten-3-ol/3-Pentanone/2-Pentanone; 2,3-Butanedione	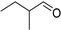	[9,13]
C4	107	1.12	(C_7_H_6_O)H^+^	Benzaldehyde	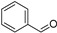	[8,9]
C5	71	1.11	(C_4_H_6_O)H^+^	2-Methyl-2-propenal	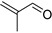	[13]
C6	149	1.11	(C_10_H_12_O)H^+^	Cuminaldehyde	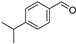	[8]
C7	59	1.11	(C_3_H_6_O)H^+^	2-Propanone	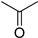	[9,13]
C8	205	1.10	(C_15_H_24_)H^+^	α-Cubebene/Copaene/Ylangene/β-Cubebene/β-Caryophyllene/Caryophyllene/Humulene/Germacrene D/α-Farnesene/(+)-δ-Cadinene/1,2,3,4,6,8a-Hexahydro-1-isopropyl-4,7-dimethylnaphthalene/α-Caryophyllene/(+)-Aromadendrene/Elemene/Bicyclosesquiphellandrene/α-Gurjunene/Valencene/Aristolene/Longifolene/δ-Elemene/β-Elemene/Eremophilene/α-Serinene/β-Cadinene/γ-Elemene/β-Guaiene/2,4-Diisopropenyl-/Gemacrene B/4,7-Dimethyl-1-isopropyl naphthalene	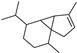	[1,4,8,10,11,13]

*^a^* Chemical formulas were predicted using the PTR-MS Viewer 3.4 software based on a library of recognized materials, that concurs with the literature as well. *^b^* Symbol “/” indicated the possible existence of more tentative identifications corresponding to one chemical formula. *^c^* One chemical structure was suggested according to the chemical formula.

## Data Availability

The original contributions presented in the study are included in the article/Appendix A, further inquiries can be directed to the corresponding authors.

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
