# Peer review of "Quality Change of Citri Reticulatae Pericarpium (Pericarps of Citrus reticulata ‘Chachi’) During Storage and Its Sex-Based In Vitro Digestive Performance"

_foods, 2024, doi:10.3390/foods13223671_

Round 1
Reviewer 1 Report
Comments and Suggestions for Authors
I reviewed the manuscript titled “Quality change of Citri Reticulatae Pericarpium
(pericarps of Citrus reticulata ‘Chachi’) during storage and its sex-based in vitro digestive performance” . the manuscript is well written. However, suggestions below may improve the quality of the manuscript.
Abstract
The research findings should be included in the abstract
Introduction
The scientific name must be in Italics
Methodology is appropriate
Results and discussion
Lines 295-299: not required
There is no limit for Tables. I suggest to include all supplementary Tables in the main text. For example, Table S3
Discussion should be improved for the section 3.1.
Figure 1 is difficult to read it
Authors studied about Citri Reticulatae Pericarpium but during the discussion authors cited many others like cabbage or mushroom etc. It is suggested to provide a discussion based on relevance.
Figure 2 is completely unreadable font size
Authors should detail about the use of PCA and heat map in methodology. Comparison parameters etc.
3.2.3. Changes in flavonoid levels during storage and accelerated spoilage: authors expressed their opinion. It is recommended to cite relevant literature for the claims they made
Figure 4: confusing figure. Too much information. It is suggested to present with two different temp separately
Please add more relevant citations to 3.3. Evaluation of in vitro digestive performance of GCP
Conclusions should be revised
References should be cross-checked
Kindly add the graphical abstract to understand the work clearly.
Author Response
Reviewer 1
Comment 1: I reviewed the manuscript titled “Quality change of Citri Reticulatae Pericarpium (pericarps of Citrus reticulata ‘Chachi’) during storage and its sex-based in vitro digestive performance”. the manuscript is well written. However, suggestions below may improve the quality of the manuscript.
Response 1: Thank you for your encouraging words and for your useful suggestions. Please see below our detailed responses.
Comment 2: Abstract- The research findings should be included in the abstract
Response 2: We have revised the abstract to offer more insights into the quantitative aspects of the work, from Lines 23-33 and marked in red.
Comment 3: Introduction- The scientific name must be in Italics
Response 3: Corrected in the revised manuscript to ensure all scientific names are presented in italics throughout the text.
Comment 4: Methodology is appropriate
Response 4: We appreciate your positive feedback on the work’s experimental design.
Comment 5: Results and discussion- Lines 295-299: not required
Response 5: As suggested, lines 295-299 have been removed.
Comment 6: There is no limit for Tables. I suggest to include all supplementary Tables in the main text. For example, Table S3
Response 6: Thank you for your suggestion regarding the inclusion of supplementary tables in the main text. In response, we have incorporated Supplementary Tables S1 and S3 as Table 1 (Line 138) and Table 2 (Line 322) in the main text, respectively. However, we decided to keep Supplementary Table S2 in the supplementary materials, as its content is not extensively discussed in the main text.
Comment 7: Discussion should be improved for the section 3.1.
Response 7: As suggested, section 3.1 has been modified to offer a more detailed discussion with additional literary support (Line 309-314).
Comment 8: Figure 1 is difficult to read it
Response 8: Thanks for your comment. Figure 1 (updated as Figure 2 in line 380 as a flow chart was added as Figure 1) has been revised to improve legibility.
Comment 9: Authors studied about Citri Reticulatae Pericarpium but during the discussion authors cited many others like cabbage or mushroom etc. It is suggested to provide a discussion based on relevance.
Response 9: Scant relevant research is available on Citri Reticulatae Pericarpium, particularly regarding respiration monitoring. Thus, we have tried to professionally discuss respiration of other fruits and vegetables, such as cabbage. In order to avoid reader confusion, we have removed “Researchers found that the total sugar contents in cabbage decreased with longer time and higher temperatures during the respiratory [30]”.
Comment 10: Figure 2 is completely unreadable font size
Response 10: We have increased the font size and image resolution and size to ensure readability. Please see it updated as Figure 3 in page 12.
Comment 11: Authors should detail about the use of PCA and heat map in methodology. Comparison parameters etc.
Response 11: Section 2.5 has been modified and elaborated, as suggested.
Comment 12: 3.2.3. Changes in flavonoid levels during storage and accelerated spoilage: authors expressed their opinion. It is recommended to cite relevant literature for the claims they made
Response 12: We have added relevant citations to support our notions in Section 3.2.3. Specifically, we have incorporated more literature in line 507 and line 522. These additions strengthen the discussion of our results.
Comment 13: Figure 4: confusing figure. Too much information. It is suggested to present with two different temp separately
Response 13: Our goal is to offer a comparative overview of findings. Hence, we think presenting two storage conditions in one figure will make it easier to see the differences. Further, we have adjusted the front size and the size of the figure 4 in page 19-20 (referring to Figure 5 in the updated version).
Comment 14: Please add more relevant citations to 3.3. Evaluation of in vitro digestive performance of GCP
Response 14: As suggested, we have incorporated additional relevant citation to strengthen Section 3.3. We have added the relevant evidence and citation in Line 560-563.
Comment 15: Conclusions should be revised
Response 15: Section 4 has been revised to better encapsulate the findings and implications of our study.
Comment 16: References should be cross-checked
Response 16: Thanks, the references have been cross-checked.
Comment 17: Kindly add the graphical abstract to understand the work clearly.
Response 17: We have added a graphical abstract at the end of the manuscript.

Reviewer 2 Report
Comments and Suggestions for Authors
In this study, the effects of the storage conditions and ageing duration on the phytochemical quality of Citri Reticulatae Pericarpium and compounds' bioaccessibility in simulated male and female digestion models were investigated. In my opinion, the topic of this study concerning determination is interesting and follows current trends in food science and technology. The abstract properly summarizes the content of the manuscript. The introduction provides a good background and clearly states the objectives of the study. References are relevant to the work and in most cases come from recent years. The experiments are rather well-planned, especially the determination of the potential bioaccessibility of phytochemicals deserves special attention. Neveertheelles, in the digestion experiment, besides naringenin, it will be valuable to provide data on the bioaccessibility for other investigated compounds (i.e. didymin, nobiletin, tangeretin, and 3,5,6,7,8,3',4'-heptamethoxyflavone). The Materials and Methods section is described in sufficient detail to reproduce experiments or necessary references are provided. Results are properly described and the modes of their presentation are clear. The discussion is supported by the results. The conclusion summarizes the most important findings. In my opinion, the manuscript meets the basic requirements of the “Foods” journal.
Author Response
Comment 1: In this study, the effects of the storage conditions and ageing duration on the phytochemical quality of Citri Reticulatae Pericarpium and compounds' bioaccessibility in simulated male and female digestion models were investigated. In my opinion, the topic of this study concerning determination is interesting and follows current trends in food science and technology.
Response 1: Thank you for acknowledging the relevance of the study and your positive feedbacks.
Comment 2: The abstract properly summarizes the content of the manuscript.
Response 2: Thank you, please note abstract has been further tweaked.
Comment 3: The introduction provides a good background and clearly states the objectives of the study.
Response 3: Thank you, please note introduction has been further tweaked.
Comment 4: References are relevant to the work and in most cases come from recent years.
Response 4: We appreciate you affirmation of the timeliness of our work and cited references.
Comment 5: The experiments are rather well-planned, especially the determination of the potential bioaccessibility of phytochemicals deserves special attention.
Response 5: We appreciate you found the experimental section clear.
Comment 6: Nevertheless, in the digestion experiment, besides naringenin, it will be valuable to provide data on the bioaccessibility for other investigated compounds (i.e. didymin, nobiletin, tangeretin, and 3,5,6,7,8,3',4'-heptamethoxyflavone).
Response 6: Indeed, Naringenin is not the sole bioactive compound, however, testing the profiles and trajectories of various antioxidants is outside the scope of the present work. We believe this is an excellent idea for a follow-up research.
Comment 7: The Materials and Methods section is described in sufficient detail to reproduce experiments or necessary references are provided.
Response 7: Thank you for the affirmation and support of our experimental design.
Comment 8: Results are properly described and the modes of their presentation are clear.
Comment 9: The discussion is supported by the results.
Comment 10: The conclusion summarizes the most important findings.
Response 8+9+10: Thank you for recognizing the clarity and legibility our work and the provided findings.
Comment 11: In my opinion, the manuscript meets the basic requirements of the “Foods” journal.
Response 11: We really appreciate your positive feedback and support for the publication of this manuscript.

Reviewer 3 Report
Comments and Suggestions for Authors
The paper “Quality change of Citri Reticulatae Pericarpium 3 (pericarps of Citrus reticulata ‘Chachi’) during storage and its sex-based in vitro digestive performance” presents valuable information regarding the bioaccessibility of some bioactive compounds and also the possibility to have bio-sensors as markers for shelf life.
The paper is well written, even if somehow is difficult to follow the main idea. I suggest a work flow to highlight the steps for the experimental design.
The authors selected the samples aging 6, 9 and 10 years and I think that there should be an explanation for choosing this range of storage time. I think that using samples from 1 to 10 years with a sampling of every 3 years would be more accurate. Moreover, comparing the obtained data, from samples of different aging, with those of fresh samples would be an aspect to be addressed in the future studies.
Author Response
Comment 1: The paper “Quality change of Citri Reticulatae Pericarpium (pericarps of Citrus reticulata ‘Chachi’) during storage and its sex-based in vitro digestive performance” presents valuable information regarding the bioaccessibility of some bioactive compounds and also the possibility to have bio-sensors as markers for shelf life.
Response 1: Thank you for your positive feedback on the manuscript's scientific merits.
Comment 2: The paper is well written, even if somehow is difficult to follow the main idea. I suggest a work flow to highlight the steps for the experimental design.
Response 2: We appreciate your suggestion and have added a workflow diagram in the methods section.
Comment 3: The authors selected the samples aging 6, 9 and 10 years and I think that there should be an explanation for choosing this range of storage time. I think that using samples from 1 to 10 years with a sampling of every 3 years would be more accurate.
Response 3: Indeed, a range from 1 to 10 years would be a rational timed-lapsed selection, however, this project opted for the detailed selection due to pragmatic considerations and availability at the time of the experiments. We have inserted our selection explanation from Line 100-105. In line with this valuable suggestion, our follow-up work is performing controlled aging of CRP. Additionally, according to the standard outlined in the Product of geographical indication - Xinhui Chenpi (DB4407/T 70-2021), GCP products from the Xinhui district must be aged locally for at least three years; peels aged for less than 3 years cannot be classified as GCP.
Comment 4: Moreover, comparing the obtained data, from samples of different aging, with those of fresh samples would be an aspect to be addressed in the future studies.
Response 4: We concur and really appreciate your recommendation.

Reviewer 4 Report
Comments and Suggestions for Authors
Dear authors, the revised manuscript is interesting. It is recommended to review the following:
Line 16: remove italic text format for of
Line 21: was to
Line 64: check type and size text format
Line 86: rewrite… (HMF) were
Line 96: remove Citrus reticulata ‘Chachi’, it was abbreviated previously
Line 98: use the abbreviature described previously
Line 106: °C
Line 113: check type and size text format
Line 128: rewrite… 2.2. Proximate Analysis of GCP
Line 131: °c
Line 131: 6 h
Line 133: insert the number of the standard procedure
Line 134: insert the reference or references [number] of the described procedures
Line 147: rewrite… 2.3. Monitoring of GCP during Storage and Accelerated Spoilage
Line 150: 25 °C
Line 150: According to the authors' guide, the name and/or surname of the authors should not appear in the text, it is only required to indicate the reference number that corresponds to the information and place it in brackets.
Line 153: °C
Line 162: °C
Line 172: check type and size text format
Line 175: rewrite… 2.3.2. Emission of VOCs during storage and accelerated spoilage
Line 177: °C
Line 185: °C
Line 194: °C
Line 195: °C
Line 215: rewrite… 2.4. Evaluation of In Vitro Digestive Performance of GCP
Line 216: rewrite… 2.4.1. Sex-based in vitro gastrointestinal…
Line 220: °C
Line 231: 2-h
Line 231: 3-h
Line 232: 2-h
Line 247: insert the temperature °C used during the centrifuging process
Line 264: align the coinage number with its respective line
Line 265: align the coinage number with its respective line
Line 278: rewrite… 2.5. Experimental Design and Data Analysis
Line 286: check type and size text format
Line 300: rewrite… 3.1. Proximate Analysis of GCP
Line 337: In the bullets in the figure, it is required to increase the thickness of the line of each bullet. These are very thin and the treatment to which the graphic information corresponds is not easily appreciated.
Line 395: °C
Line 397: °C
Line 398: °C
Line 404: rewrite… Table 1. Characteristic VOCs emitted from….
Line 405: °C
Page 17: line number continuity is lost
Note: correct °C through the manuscript
Page 19: rewrite… 3.3. Evaluation of In Vitro Digestive Performance of GCP
Page 20: The upper graph seems to be the information behind the other figure within the same graph.
Note: Check the correct text format for each of the references
Author Response
Comment 1: Dear authors, the revised manuscript is interesting. It is recommended to review the following:
Response 1: We appreciate your recommendations and have made the revisions as suggested.
Comment 2: Line 16: remove italic text format for of
Response 2: corrected in the revised manuscript.
Comment 3: Line 21: was to
Response 3: We have corrected the wording in Line 21 to address the issue you pointed out.
Comment 4: Line 64: check type and size text format
Response 4: We have reviewed the text format in Line 64.
Comment 5: Line 86: rewrite… (HMF) were
Response 5: We have rewritten Line 86 to improve clarity.
Comment 6: Line 96: remove Citrus reticulata ‘Chachi’, it was abbreviated previously
Response 6: Corrected in the revised manuscript.
Comment 7: Line 98: use the abbreviature described previously
Response7: While pericarp of Citrus reticulata ‘Chachi’ is abbreviated as GCP, we would like to clarify that in Line 108, Citrus reticulata ‘Chachi’ refers specifically to the type of orange rather than its pericarp. Therefore, we believe it would be more appropriate to keep the full name to avoid confusion.
Comment 8: Line 106: °C
Response 8: Formatting issue corrected.
Comment 9: Line 113: check type and size text format
Response 9: Done.
Comment 10: Line 128: rewrite… 2.2. Proximate Analysis of GCP
Response 10: We have revised Line 128 to “Proximate Analysis of GCP.” We have also made the necessary corrections throughout the manuscript.
Comment 11: Line 131: °c
Response 11: Done
Comment 12: Line 131: 6 h
Response 12: We have revised “6 hr” to “6 h” in Line 131 to align with standard formatting conventions and have checked the rest of the manuscript as well.
Comment 13: Line 133: insert the number of the standard procedure
Response 13: We have revised the sentence in Line 145 to include the standard procedure number (No. 2540).
Comment 14: Line 134: insert the reference or references [number] of the described procedures
Response 14: In Line 134, the determination methods refer to the State Standard of the People's Republic of China (NY/T 2017-2011, GB/T 15672-2009, GB 5009.5-2016, GB 5009.268-2016) by Convinced-Test Technology Co., LTD (China), as outlined in Lines 155–156. Citing official standards like these can be challenging since they’re often not available online or widely cited in the same way as academic articles. Therefore, we have only provided their names in the manuscript for clarity. Thanks for your understanding.
Comment 15: Line 147: rewrite… 2.3. Monitoring of GCP during Storage and Accelerated Spoilage
Response 15: Done.
Comment 16: Line 150: 25 °C
Response 16: Done.
Comment 17: Line 150: According to the authors' guide, the name and/or surname of the authors should not appear in the text, it is only required to indicate the reference number that corresponds to the information and place it in brackets.
Response 17: We appreciate your feedback and have revised the text to ensure compliance with the authors' guide, in Line 164-166.
Comment 18: Line 153: °C
Response 18: Done.
Comment 19: Line 162: °C
Response 19: Done.
Comment 20: Line 172: check type and size text format
Response 20: Done.
Comment 21: Line 175: rewrite… 2.3.2. Emission of VOCs during storage and accelerated spoilage
Response 21: We have revised Line 175.
Comment 22: Line 177: °C
Response 22: Done.
Comment 23: Line 185: °C
Response 23: Done.
Comment 24: Line 194: °C
Response 24: Done.
Comment 25: Line 195: °C
Response 25: Done.
Comment 26: Line 215: rewrite… 2.4. Evaluation of In Vitro Digestive Performance of GCP
Response 26: Done.
Comment 27: Line 216: rewrite… 2.4.1. Sex-based in vitro gastrointestinal…
Response 27: Done.
Comment 28: Line 220: °C
Response 28: Done.
Comment 29: Line 231: 2-h
Response 29: Done.
Comment 30: Line 231: 3-h
Response 30: Done.
Comment 31: Line 232: 2-h
Response 31: Done.
Comment 32: Line 247: insert the temperature °C used during the centrifuging process
Response 32: We have clarified centrifugation was done at ambient temperature in Line 260.
Comment 33: Line 264: align the coinage number with its respective line
Response 33: We regret any inconvenience from this technical issue of the journal submission system. We have double-checked and ensured that the coinage number is aligned correctly with its respective line in the manuscript.
Comment 34: Line 265: align the coinage number with its respective line
Response 34: we have aligned the coinage number with its respective line.
Comment 35: Line 278: rewrite… 2.5. Experimental Design and Data Analysis
Response 35: Done.
Comment 36: Line 286: check type and size text format
Response 36: We have reviewed Line 286 and ensured that the type and size of the text are consistent.
Comment 37: Line 300: rewrite… 3.1. Proximate Analysis of GCP
Response 37: Done.
Comment 38: Line 337: In the bullets in the figure, it is required to increase the thickness of the line of each bullet. These are very thin and the treatment to which the graphic information corresponds is not easily appreciated.
Response 38: Thanks for your suggestion. We have tried to increase the line thickness, but lines of each sample might overlap, making it harder to read. Therefore, we have increased the size of the front and the size of each figure to make it clear and easier for reading, please see in Figure 2 (updated) in Page 10-11.
Comment 39: Line 395: °C
Response 39: Done.
Comment 40: Line 397: °C
Response 40: Done.
Comment 41: Line 398: °C
Response 41: Done.
Comment 42: Line 404: rewrite… Table 1. Characteristic VOCs emitted from….
Response 42: Thanks for your notice. We have only retained the abbreviation “VOCs” and removed the full name.
Comment 43: Line 405: °C
Response 43: Done.
Comment 44: Page 17: line number continuity is lost
Response 44: We apologize and have revised the line number and ensured the consistency of line numbering throughout the manuscript. Thank you for bringing this to our attention.
Comment 45: Note: correct °C through the manuscript
Response 45: We have checked and revised the formatting of "°C" throughout the manuscript to ensure it is correctly displayed.
Comment 46: Page 19: rewrite… 3.3. Evaluation of In Vitro Digestive Performance of GCP
Response 46: Done.
Comment 47: Page 20: The upper graph seems to be the information behind the other figure within the same graph.
Response 47: We have adjusted the figure's location to ensure that the information does not overlap with the other figure.
Comment 48: Note: Check the correct text format for each of the references
Response 48: We have checked and revised the text format for each of the references
